# The Importance of Strain (Preorganization) in Beryllium Bonds

**DOI:** 10.3390/molecules25245876

**Published:** 2020-12-11

**Authors:** Ibon Alkorta, José Elguero, Josep M. Oliva-Enrich, Manuel Yáñez, Otilia Mó, M. Merced Montero-Campillo

**Affiliations:** 1Instituto de Química Médica (CSIC), Juan de la Cierva, 3, E-28006 Madrid, Spain; iqmbe17@iqm.csic.es; 2Instituto de Química-Física “Rocasolano” (CSIC), Serrano, 119, E-28006 Madrid, Spain; 3Departamento de Química, Facultad de Ciencias, Módulo 13, and Institute of Advanced Chemical Sciences (IadChem), Campus de Excelencia UAM-CSIC, Universidad Autónoma de Madrid, Cantoblanco, E-28049 Madrid, Spain; otilia.mo@uam.es (O.M.); mm.montero@uam.es (M.M.M.-C.)

**Keywords:** *ortho*-carboranes, beryllium bond, Lewis acid-Lewis base complexes, LMOEDA

## Abstract

In order to explore the angular strain role on the ability of Be to form strong beryllium bonds, a theoretical study of the complexes of four beryllium derivatives of *ortho*
*closo*-carboranes with eight molecules (CO, N_2_, NCH, CNH, OH_2_, SH_2_, NH_3_, and PH_3_) acting as Lewis bases has been carried out at the G4 computational level. The results for these complexes, which contain besides Be other electron-deficient elements, such as B, have been compared with the analogous ones formed by three beryllium salts (BeCl_2_, CO_3_Be and SO_4_Be) with the same set of Lewis bases. The results show the presence of large and positive values of the electrostatic potential associated to the beryllium atoms in the isolated four beryllium derivatives of *ortho*-carboranes, evidencing an intrinsically strong acidic nature. In addition, the LUMO orbital in these systems is also associated to the beryllium atom. These features led to short intermolecular distances and large dissociation energies in the complexes of the beryllium derivatives of *ortho*-carboranes with the Lewis bases. Notably, as a consequence of the special framework provided by the *ortho*-carboranes, some of these dissociation energies are larger than the corresponding beryllium bonds in the already strongly bound SO_4_Be complexes, in particular for N_2_ and CO bases. The localized molecular orbital energy decomposition analysis (LMOEDA) shows that among the attractive terms associated with the dissociation energy, the electrostatic term is the most important one, except for the complexes with the two previously mentioned weakest bases (N_2_ and CO), where the polarization term dominates. Hence, these results contribute to further confirm the importance of bending on the beryllium environment leading to strong interactions through the formation of beryllium bonds.

## 1. Introduction

In the last years, the importance of non-covalent interactions (NCI) in a number of fields has been highlighted. Parallel to these activities, new types of NCIs have been described and named using the Periodic Table column of the atom of the Lewis acid involved in the interaction [1,2]. Thus, aerogen [3], halogen [4,5], chalcogen [6,7,8,9], tetrel [10,11], triel [12,13], spodium [14], regium or coinage-metal [15,16,17,18], alkaline-earth [19,20], and alkaline bonds [21] have been described, which are associated with atoms of columns 18-11 and 2-1, respectively. Beryllium bonds [22,23] are a subfamily of the alkaline-earth NCIs that show very strong binding energies and profound changes in the geometric and electronic properties of the systems involved.

Icosahedral *closo*-heteroboranes are a family of compounds with 12 atoms forming the central framework. The parent compound is the very stable dianion (B_12_H_12_)^2−^. Derivatives with two carbon atoms in the skeleton are neutral and named *closo*-carboranes. They have been used from potential scaffolds in the design of new drugs, as a catalyst and for energy storage [24,25]. Among the different *closo*-carboranes with metallic replacement in the cage, in 1971, Popp and Hawthorne [26] described the single case in the literature with a beryllium atom, 3-beryl-1,2-dicarbacloso-dodecaborane(12). The experimental evidences indicated that this molecule forms complexes with diethylether or trimethylamine in the presence of these bases (Figure 1).

Strained systems have been shown to react more easily than other related non-strained systems. In some cases, this effect could be due to a pre-organization of the system to evolve more easily. For instance, the hydrogen bond donor ability of C(*sp*^3^)-H groups increases significantly when they are located in three membered rings [27], the hydrolysis of amides have lower barriers when they are highly twisted [28] and strained beryllium derivatives become excellent H_2_ acceptors [29,30].

In this article, we study the complexes formed by the particular set of beryl-1,2-dicarbacloso-dodecaboranes, where the metallic replacement introduces a typical electron deficient atom, Be, in a rather strained environment, involving also other electron-deficient atoms, such as B. More precisely, the structure, stability, and bonding characteristics of the complexes between four beryl-dicarbacloso-dodecaborane isomers and eight different Lewis bases (CO, N_2_, NCH, CNH, OH_2_, SH_2_, NH_3_, and PH_3_) have been studied at G4 computational level. These compounds are shown in Figure 2, where four different isomers, **A**–**D**, corresponding to the different position of the C-C linkage within the cage. In addition, the complexes formed between the same set of bases and Cl_2_Be, CO_3_Be, and SO_4_Be salts, where Be is not attached to electron-deficient elements and has different degrees of strain, have been calculated and compared with the results obtained for the carboranes.

## 2. Computational Methods

The geometry of the isolated molecules and complexes was optimized with the Gaussian-4 (G4) [31] formalism at the B3LYP/6-31G (2df,p) level of theory. Frequency calculations at B3LYP/6-31G (2df,p) level were used to confirm that the complexes correspond to energy minima. Thermochemical corrections were also computed with the G4 formalism and the Gaussian-16 software [32].

The dissociation enthalpies of the complexes were calculated as the sum of the values of the isolated monomers minus that of the complex calculated at G4 level. The deformation energy was obtained as the G4 electronic energy difference between the molecules in the geometry of the complex and in their minima conformation, as isolated molecules.

The molecular electrostatic potential of the isolated molecules was calculated at B3LYP/6-31(2df,p) level, analyzed on the 0.001 au electron density isosurface with the Multiwfn program [33], and represented with the Jmol program [34].

The electron density distribution was analyzed within the quantum theory of Atoms in Molecules (QTAIM) methodology [35,36] and the AIMAll program [37]. The critical points of the electron density correspond to maxima (3,–3) associated to the location of the atom, two types of saddle points, (3,–1) and (3,+1), associated with the presence of bond and ring critical points, respectively, and minima (3,+3), associated to cage critical points.

The localized molecular orbital energy decomposition (LMOEDA) method [38] at B3LYP/6-31G(2df,p) level was carried out in order to analyze the importance of the different energy terms in the interaction energy. Based on this method, the interaction energy can be decomposed in a repulsive term and several attractive terms (electrostatic, exchange, polarization and dispersion energies). The electrostatic term describes the classical Coulomb interaction. The repulsion and exchange terms are associated to Pauli’s exclusion principle. The polarization term is caused by the relaxation of the electronic distribution in the complex. The dispersion term arises from the electron correlation. These calculations were carried out with the GAMESS program [39].

## 3. Results and Discussion

### 3.1. Isolated Be-Containing Lewis Acids

In order to understand and rationalize how the different dodecaboranes may interact with the selected set of Lewis bases, we will firstly analyze the isolated dodecaboranes. The four isomers with general formula B_9_H_11_C_2_Be considered in this article show an icosahedral shape (Figure 3). Only isomer **A** has been described in the literature [26]. The relative enthalpy of the four isomers is listed in Table 1. The most stable isomer **D** is the one with the maximum distance between the carbon and beryllium atoms, whereas isomer **A**, which exhibits the shortest distance between those atoms, lies 18 kJ·mol^−1^ higher in energy (see Table 1). The other two isomers, **C** and **B**, show relative energies of 29 and 35 kJ·mol^−1^, respectively. Thus, no clear relationship is observed between the relative stability of the isomers and the distribution of the atoms in the icosahedral structure of the molecule.

The electrostatic potential on the 0.001 au electron density isosurface of the molecules (see Table 1) shows its maximum value, V_s, max_, in the proximity of the beryllium atom (see Figure 4a for isomer **A**, as a suitable example. Appendix A of the Appendix A shows the same information for isomers **B**–**D**). The largest value is obtained for isomer **A** (669 kJ·mol^−1^), in which the Be atom is directly linked to both carbon atoms. The following value (626 kJ·mol^−1^) corresponds to isomer **B**, in which Be is directly bonded to only one of the two carbon atoms. The electrostatic potential for isomers **C** and **D** is decreasingly smaller. However, for isomer **C**, this potential is higher than for isomer **D**, because in the former, two of the B atoms attached to Be are bonded to the C atoms, whereas in isomer **D**, only one is attached to a C atom. For comparative purposes, the values corresponding to Cl_2_Be, CO_3_Be, and SO_4_Be have also been included in the same table. The values obtained for SO_4_Be (938 kJ·mol^−1^) and CO_3_Be (907 kJ·mol^−1^) are larger than those obtained for the B_9_H_11_C_2_Be isomers, reflecting the strong electron withdrawing power of the sulfate and carbonate groups, while in the case of Cl_2_Be (141 kJ·mol^−1^), the value of V_s, max_ is very small as compared to the rest of the beryllium derivatives considered here.

As regards to the molecular orbitals, the LUMO orbital is mostly associated to the beryllium atom in all B_9_H_11_C_2_Be isomers (see Figure 4b for isomer **A** and Appendix A for isomers **A**–**D**). It is interesting to note that the energy of the LUMO for all these compounds is negative (see Appendix A of the Appendix A), in agreement with the electron deficient character of Be. Interestingly, these energies, in absolute value, follow the trend **A** > **B** > **C** > **D**, which is the same as the trend observed for the values of the V_s, max_ of MESP (see Table 1). In fact, there is a reasonably good linear correlation between both sets of values (see Appendix A). Thus, from both an electrostatic and orbital perspective, the most acidic regions of these molecules are associated with the location of the beryllium atom, and the intrinsic acidities of these compounds should follow **A** > **B** > **C** > **D**, which implies that the most stable isomer is the least reactive and the second most stable isomer is the most reactive one. We will see later on that the interaction energies between these four isomers and the different Lewis bases considered in this study follow exactly the same trend, **A** > **B** > **C** > **D**.

As a measure of the geometrical strain of the beryllium atom in each of the B_9_H_11_C_2_Be isomers, the distance between this atom and the plane defined by the five atoms attached to it has been calculated. Ideally, a beryllium atom in the same plane to those of the surrounding atoms should not have strain, but as the distance to the plane increases, the strain should increase too. More importantly, the calculated distance values (1.030, 1.014, 1.010, and 1.000 Å for isomers **A**–**D**, respectively) have been found to correlate linearly (R^2^ = 0.99) with the calculated V_s, max_, a clear indication that changes in the strain at the Be center are closely related with its electrophilicity.

The molecular graphs of the four isomers are depicted in Figure 5, and show the presence of only three bond critical points (BCPs) between the Be atom and the surrounding atoms for **A**, **B**, and **D** isomers and five BCPs in **C**. However, in the latter case, two of the BCPs are in close proximity of an RCP, which indicates the possibility of their disappearance upon some external perturbation. In any case, the Be-C and Be-B BCPs obtained for all the isomers show a small value of the electron density (between 0.079 and 0.075 au), as should be expected for bonds involving electron-deficient atoms, but with negative values of the energy density, which indicates that the interactions have some covalent character. The B-B and the C-C BCPs in these systems show electron densities between 0.123 and 0.109 au for the B-B linkages and around 0.18 au for the C-C ones. For both kinds of bonds, the total energy density is negative, as should be expected for covalent interactions.

### 3.2. B_9_H_11_C_2_Be: Lewis Base Complexes

The structures of the most stable complexes formed by the interaction of the Lewis bases under consideration with the beryllium atoms of the four B_9_H_11_C_2_Be isomers **A**–**D** have been located. As an illustration, we show in Figure 6 the molecular graphs of four suitable examples, but a complete information on the structures of these complexes is shown in Appendix A of the Appendix A.

The molecular graph of the complexes presents a unique BCP between the two interacting molecules. The BCPs are always located closer to the beryllium atoms than to the base, as expected due to the smaller size and more electropositive character of the beryllium. The electron density at this intermolecular BCP ranges from 0.038 to 0.080, but what is more important, for a given Lewis base, the value of the electron density at this BCP (see Appendix A) follows the sequence **A** > **B** > **C** > **D** (except for the CO and CNH complexes), confirming the picture obtained from the energy of the LUMO and the values of the V_s, max_, which indicate again that the most stable isomer is the least reactive and the less stable isomer **A** the most reactive one. Additionally, in all cases, the values of the Laplacian (between 0.09 and 0.57 au) are positive, consistent with a dominant electrostatic interaction.

From the values in Table 2, it can be seen that the intermolecular distances between the Be atom and the interacting atom of the Lewis Base are always very short, pointing to a rather strong interaction. For those cases where the basic site of the Lewis base belongs to the first row of the periodic table (C, N, and O), the interatomic distances are between 1.60 and 1.88 Å, while for the second-row basic sites (S and P), the distances range between 2.11 and 2.28 Å.

Exponential relationships are obtained between the electron densities at the base-Be BCP and the Be-base interatomic distances, as long as the data are grouped based on the nature of the basic center of the Lewis base (Appendix A) in agreement with other interactions [40,41,42]. In the case of the Laplacian, the correlation with the interatomic distance is independent of the atom interacting with Be (Appendix A). Importantly, the total energy density is negative for all the complexes with CNH, CO, NH_3_, SH_2_, and PH_3_, indicating a certain covalent character of the interaction. Nonetheless, for all complexes with N_2_ and OH_2_ and those of NCH with the beryllium carboranes, positive values of the energy density are obtained.

In order to gain some insight on these findings, we have used the LMOEDA energy partition method, which permits to analyze the contribution of the different energy terms to the dissociation energies (Appendix A). In the **A**–**D** complexes, the most important term is the repulsion energy, which is overcome by the sum of the attractive terms (electrostatic, exchange, polarization, and dispersion). Among the attractive terms, the two most important ones are the electrostatic and the polarization terms. The electrostatic contribution is dominant and increases with the binding energy for stronger complexes, i.e., (SH_2_, PH_3_, NCH, OH_2_, CNH, and NH_3_), with a top of 51% for NH_3_ complexes. The opposite happens with the polarization, which is the most important term in the two weakest complexes (N_2_ and CO), with contributions between 44 and 42% to the total attractive interaction. The two less important attractive terms, exchange and dispersion, have contributions between 15–21% and 6–9% of the sum of the attractive terms, respectively. The LMOEDA partition results for CO_3_Be, SO_4_Be, and Cl_2_Be complexes follow similar trends, except for an increment of the contribution of the exchange energy in the Cl_2_Be (between 20 and 24%) and a reduction of the polarization term (between 26 and 36%).

From this analysis, it is easy to justify that complexes with N_2_ exhibit positive values of the energy density, because as mentioned above, the interaction is dominated by dispersion and polarization interactions. However, a positive energy density for water complexes is unclear.

It is also worth noting that the intermolecular Be-Lewis base distances for the B_9_H_11_C_2_Be-base complexes of isomer **A** are highly correlated (R^2^ > 0.99) with those of the other three isomers, **B**–**D** (Appendix A). The quality of the correlation drops slightly when the values of the complexes with CO_3_Be and SO_4_Be are compared with those from the **A**–**D** isomers (R^2^ between 0.98 and 0.95) and even more with the Cl_2_Be complexes (R^2^ between 0.94 and 0.92). Indeed, although the values in Table 2 clearly show that the distances for the B_9_H_11_C_2_Be-containing complexes are similar to those found for the complexes with CO_3_Be, SO_4_Be, and Cl_2_Be, the differences are not always of the same sign. In all cases, the longest intermolecular distances for a given Lewis base are obtained for the Cl_2_Be complexes, save for the complex with OH_2_, with shorter distance as compared to complexes with the **B**, **C**, and **D** isomers, likely due to the effect of secondary interactions. The peculiar behavior observed for the Cl_2_Be complexes is a direct consequence of the deformation changes upon complex formation. In general, whereas these deformations are small for the Be compounds considered, Cl_2_Be is a clear exception, because a significant change in the Cl-Be-Cl angle is observed. Indeed, the Cl-Be-Cl angle in the complexes ranges between 135° and 146°, whereas the isolated structure is linear the isolated structure is linear.

The dissociation enthalpies of the complexes are gathered in Table 3, and the values of Δ*G* are listed in Appendix A of the SI. The dissociation enthalpies range between 6.1 kJ·mol^−1^ for Cl_2_Be:N_2_ to 256.1 kJ·mol^−1^ for the SO_4_Be:NH_3_ complex. For a given Lewis base, the smallest values are always obtained in the BeCl_2_ complexes, again a consequence, as we will see later, of the large deformation effects in these particular complexes, while the largest ones are found in the complexes with SO_4_Be or with isomer **A**. In the isomers of B_9_H_11_C_2_Be, the dissociation enthalpies decrease as the distance between the Be and carbon atoms increases; thus, the largest values for a given base are obtained in the complex with isomer **A**, while the smallest in those with isomer **D**. In fact, excellent correlations (R^2^ > 0.99) are obtained between the strain parameter values calculated for the isolated B_9_H_11_C_2_Be molecules and the dissociation enthalpy for each Lewis base, with the exception of N_2_ complexes. In all systems studied here for a given Lewis acid, the smallest dissociation enthalpy is always found in the N_2_ complexes, while the largest one is found in the NH_3_ complexes.

As for the intermolecular distances, the dissociation enthalpies for the **A**–**D** complexes are highly correlated among each other (R^2^ > 0.99, see Appendix A). Again, the correlations are slightly worse when the dissociation enthalpies for the **A**–**D** complexes are compared with those of the CO_3_Be- and SO_4_Be-containing complexes (R^2^ between 0.98 and 0.92) and even worse when dealing with Cl_2_Be complexes (R^2^ between 0.93 and 0.90).

Interestingly, even though the largest values of the V_s, max_ (Table 1) in the isolated beryllium derivatives correspond to SO_4_Be followed by CO_3_Be, the dissociation energies in several cases of isomer **A** are larger than those for SO_4_Be and CO_3_Be, an indication that using only MESP as a binding energy predictor is not enough for these complexes. Although the electrostatic interactions are a strong component of the stabilization of the complexes, other interaction terms, such as polarization and dispersion, play an important role in defining the correct stability trends. Another important issue to be considered is the deformation penalty when the complex is formed. The values obtained are listed in Table 4. Clearly, the complexes of Cl_2_Be have larger distortion energies (between 19 and 55 kJ·mol^−1^), which is one of the factors that contributes to the relatively low stabilization enthalpies calculated for such complexes. For the remaining complexes, the deformation energy values are smaller than 13 kJ·mol^−1^.

The dissociation energy of the complexes has been analyzed using a model proposed originally by Legon and Miller (Equation (1)) [43] that relates these energies with a nucleophilic parameter characterizing the bases, *N_b_*, and an electrophilic parameter characterizing the Lewis acids, *E_a_* [19,44,45]:(1)$$D_{e}=c\cdot \;N_{b}\cdot E_{a}$$
where the *c* constant has a value of 1.0 kJ·mol^−1^ to maintain the units of the equation.

In a first attempt, the 56 complexes have been used to obtain 7 *E_a_*’s (because in our analysis we include the four **A**–**D** isomers of the beryllium *ortho*-carboranes and the three Be salts) and 8 *N_b_*’s (corresponding to the 8 Lewis base considered) with Equation (2). The 7 LA and 8 LB have been fitted:
(2)$$D_{e}=c\cdot \left(\overset{8}{\underset{i=1}{\sum }}x_{i}\cdot N_{bi}\right)\cdot \left(\overset{7}{\underset{j=1}{\sum }}x_{j}\cdot E_{aj}\right)$$
where the values of *x_i_* and *x_j_* are 1.0 when the corresponding Lewis base or Lewis acid is present in the complex, and 0.0 otherwise.

The fitted values of *N_b_* and *E_a_* are listed in Table 5. The results with all the complexes (Model 1) clearly show that the complexes with Cl_2_Be have the largest deviations between the calculated and fitted values (Figure 7). Removing the Cl_2_Be complexes (Model 2), the statistics improve (larger R^2^ and smaller SD). These results indicate that the electrophilicities of the bases follow the order: SO_4_Be > **A** > **B** > CO_3_Be > **C** > **D** > Cl_2_Be.

The *N_b_* values for the LB in this set of complexes are larger than those derived for complexes of simple beryllium derivatives [19], but they appear in the same order as the ones found here: NH_3_ > OH_2_ > NCH > PH_3_ > CO (only those cases common to the two studies are indicated).

## 4. Conclusions

A theoretical study of four beryllium derivatives of *ortho*-carboranes, one of them known experimentally, was carried out at G4 ab initio computational level. The isolated molecules show a positive region of the electrostatic potential close to the beryllium atom and the LUMO orbital in the same region. Consistently, the Be site is the acidic center when these compounds interact with eight molecules (CO, N_2_, NCH, CNH, OH_2_, SH_2_, NH_3_, and PH_3_) acting as Lewis bases. In our survey, we have also included complexes of these same bases with three additional beryllium compounds with a different degree of strain (SO_4_Be, CO_3_Be, and Cl_2_Be). In all cases, strong complexes are formed between the beryllium derivatives and the Lewis bases reaching dissociation energies of 241 kJ mol^−1^, demonstrating the importance of the strain to lead to strong beryllium bonds. Interestingly, the interaction enthalpies of some of the beryllium *ortho*-carboranes are larger than the ones calculated for complexes with SO_4_Be. The strain around the beryllium atom has been shown to linearly correlate with the value of the positive region of the electrostatic potential and the dissociation energy values. The LMOEDA energy partition shows that among the attractive terms associated to the dissociation energy, the electrostatic one is the dominant one, being up to 51% of the sum of attractive terms except for the complexes with the two weakest bases (N_2_ and CO) where the polarization is dominant. The latter cases are specifically those for which *ortho*-carboranes led to even stronger beryllium bonds than previously reported values.

## Figures and Tables

**Figure 1 molecules-25-05876-f001:**
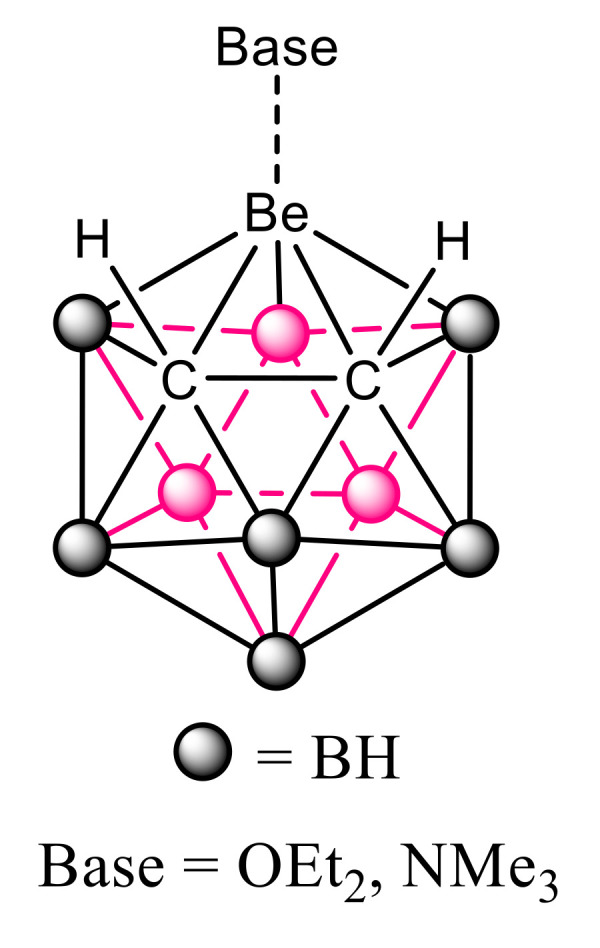
The two complexes described for the 3-beryl-l,2-dicarbacloso-dodecaborane(12) [26].

**Figure 2 molecules-25-05876-f002:**
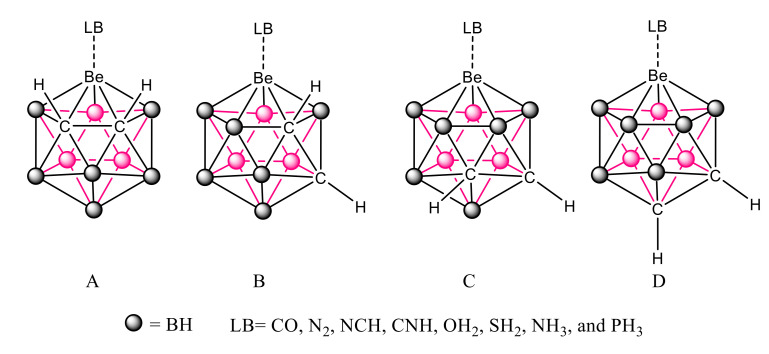
Schematic representation of the complexes between the four beryl-dicarbacloso-dodecaborane isomers (**A**–**D**) and the Lewis Bases considered in the article.

**Figure 3 molecules-25-05876-f003:**
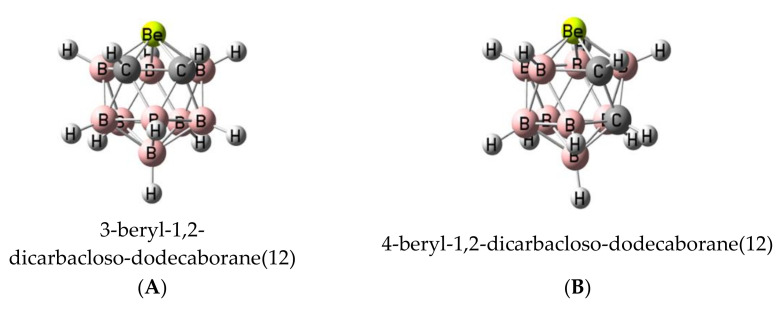
Optimized structure of the four B_9_H_11_C_2_Be isomers considered in this article with the IUPAC nomenclature.

**Figure 4 molecules-25-05876-f004:**
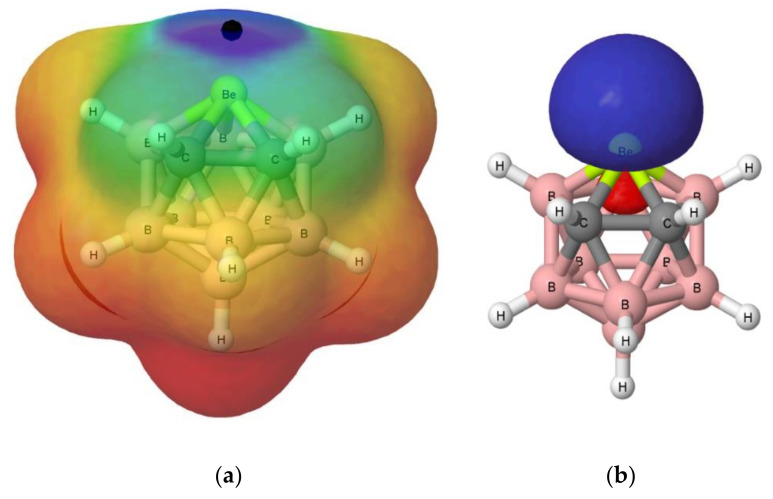
(**a**) Molecular electrostatic potential (MESP) on the 0.001 au electron density of B_9_H_11_C_2_Be (**a**). Regions with MESP > 0.1 au are indicated with blue color and MESP < −0.015 au with red color. The location of the V_s, max_ is indicated with a black sphere; (**b**) shows the LUMO orbital for the same species.

**Figure 5 molecules-25-05876-f005:**
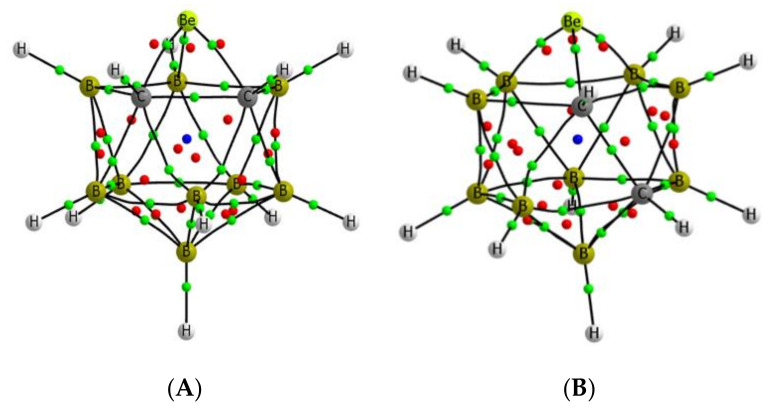
Molecular graph for the four isomers of the beryllium *ortho*-carboranes investigated. The location of the bond, ring, and cage critical points is indicated with green, red, and blue dots, respectively. **A**–**D** the same as in Figure 2.

**Figure 6 molecules-25-05876-f006:**
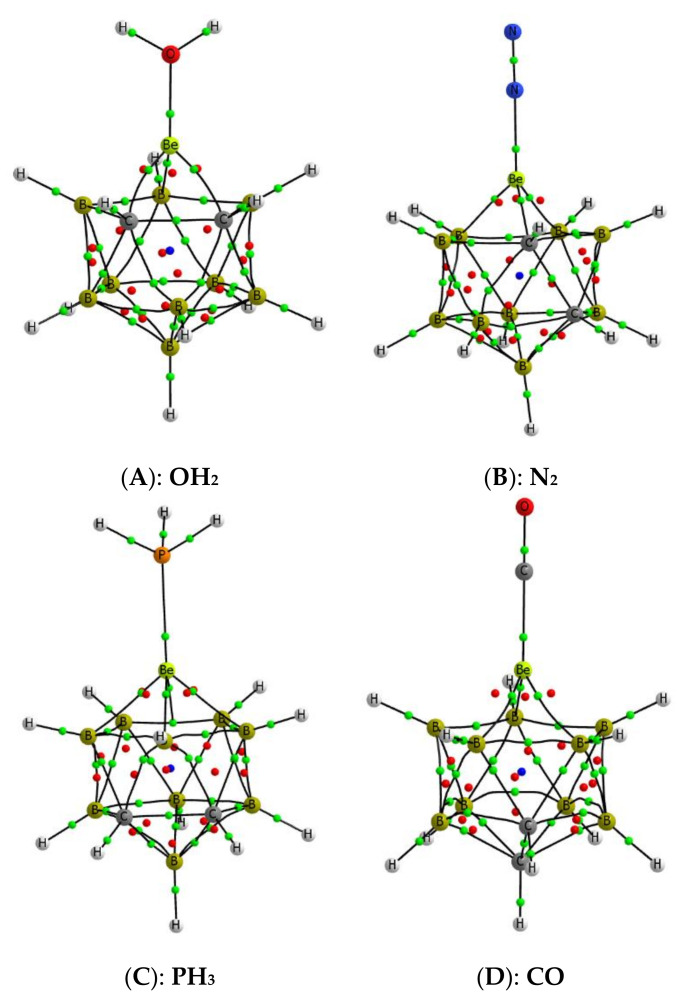
Molecular graph of some complexes. The location of the bond, ring, and cage critical points is indicated with green, red, and blue dots, respectively. **A**–**D** the same as in Figure 2.

**Figure 7 molecules-25-05876-f007:**
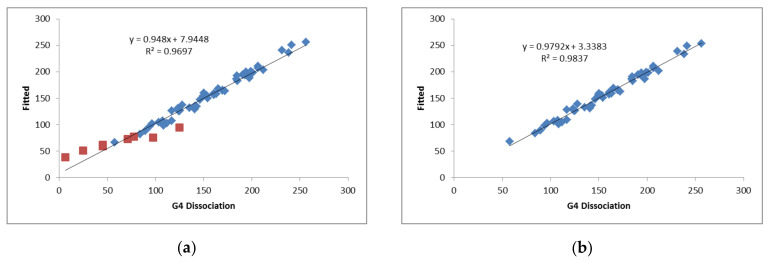
G4 Dissociation enthalpies vs. fitted values (kJ mol^−1^). (**a**) Model 1: All the complexes with the Cl_2_Be indicated with red squares. (**b**) Model 2: All the complexes except those involving Cl_2_Be.

**Table 1 molecules-25-05876-t001:** Relative enthalpies, ΔH_rel_ (kJ·mol^–1^), of the four isomers of B_9_H_11_C_2_Be, with respect to isomer **D** and V_s,max_ (kJ·mol^–1^). The V_s,max_ values of CO_3_Be, SO_4_Be, and Cl_2_Be are also included.

	ΔH_rel_	V_s, max_
B_9_H_11_C_2_Be (**A**)	17.7	669.4
B_9_H_11_C_2_Be (**B**)	34.7	625.9
B_9_H_11_C_2_Be (**C**)	29.2	605.0
B_9_H_11_C_2_Be (**D**)	0.0	575.3
CO_3_Be		906.7
SO_4_Be		938.5
Cl_2_Be		141.0

**Table 2 molecules-25-05876-t002:** Intermolecular distance (Å) between the Be atom and the basic site of the Lewis bases.

Base\Acid	A	B	C	D	CO_3_Be	SO_4_Be	Cl_2_Be
N_2_	1.692	1.690	1.688	1.691	1.724	1.729	1.884
CO	1.735	1.729	1.722	1.725	1.794	1.802	1.874
SH_2_	2.127	2.125	2.134	2.137	2.118	2.113	2.250
PH_3_	2.140	2.138	2.137	2.140	2.167	2.164	2.278
NCH	1.660	1.661	1.663	1.665	1.671	1.670	1.730
OH_2_	1.641	1.646	1.659	1.659	1.603	1.597	1.642
CNH	1.736	1.732	1.728	1.730	1.770	1.771	1.818
NH_3_	1.708	1.712	1.719	1.720	1.697	1.691	1.739

**Table 3 molecules-25-05876-t003:** Dissociation enthalpies (kJ·mol^−1^) calculated at G4 computational level.

Base\Acid	A	B	C	D	CO_3_Be	SO_4_Be	Cl_2_Be
N_2_	111.8	108.6	83.8	57.5	93.8	103.6	6.1
CO	142.9	140.6	116.7	89.6	117.1	127.7	24.7
SH_2_	160.9	154.5	124.2	96.7	146.5	160.6	45.2
PH_3_	169.6	163.4	135.3	107.6	150.0	165.3	45.2
NCH	193.3	184.0	153.0	125.3	185.1	201.0	71.1
OH_2_	194.3	184.8	150.3	123.3	197.4	212.0	97.3
CNH	206.7	198.9	169.5	141.1	190.6	206.5	77.5
NH_3_	241.2	231.6	199.2	171.9	238.6	256.1	124.7

**Table 4 molecules-25-05876-t004:** Deformation energy (kJ·mol^−1^) due to the complex formation.

	A:LB ^a^		B:LB		C:LB		D:LB		CO_3_Be:LB		SO_4_Be:LB		Cl_2_Be:LB	
Base	BeD ^b^	LB	BeD	LB	BeD	LB	BeD	LB	BeD	LB	BeD	LB	BeD	LB
N_2_	2.3	0.0	2.4	0.0	2.3	0.0	2.5	0.0	1.9	0.1	0.2	0.1	28.4	0.1
CO	3.0	0.2	3.0	0.1	2.8	0.1	3.1	0.1	2.9	0.6	0.9	0.8	34.6	0.3
SH_2_	4.2	0.3	4.2	0.3	3.9	0.2	4.2	0.2	3.2	0.5	1.9	0.5	38.4	0.4
PH_3_	4.9	7.5	4.8	6.8	4.4	6.0	4.7	5.8	4.1	8.7	2.8	9.7	41.1	5.4
NCH	5.6	0.7	5.7	0.7	5.4	0.7	5.9	0.7	4.8	1.0	3.1	1.1	48.1	0.8
OH_2_	4.7	1.1	4.9	1.1	4.9	0.9	5.3	1.0	3.7	2.5	2.3	2.6	45.1	3.9
CNH	5.5	1.6	5.4	1.4	5.2	1.2	5.6	1.2	5.1	2.1	3.3	2.4	48.0	1.5
NH_3_	7.1	0.0	7.2	0.0	7.0	0−.1	7.5	0−.1	5.3	0.1	4.0	0.1	54.8	0.3

^a^ LB stands for Lewis Base. ^b^ BeD stands for Beryllium derivative.

**Table 5 molecules-25-05876-t005:** Statistical values and fitted parameters.

	Model 1	Model 2
	All	Without BeCl_2_
R^2^	0.96	0.98
SD (kJ mol^−1^)	9.9	5.9
	*E_a_* values	
Cl_2_Be	5.5	
CO_3_Be	13.8	13.2
SO_4_Be	14.9	14.3
A	14.6	14.0
B	14.1	13.5
C	11.7	11.2
D	9.6	9.2
	*N_b_* values	
N_2_	7.0	7.5
CO	9.2	9.7
SH_2_	10.7	11.2
PH_3_	11.2	11.8
NCH	13.2	13.8
OH_2_	13.7	14.1
CNH	14.1	14.7
NH_3_	17.1	17.7

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
