# Peer review of "The Importance of Strain (Preorganization) in Beryllium Bonds"

_molecules, 2020, doi:10.3390/molecules25245876_

Round 1

Reviewer 1 Report

In this paper, the berlllium bonds in four beryllium derivatives of ortho-carboranes were explored theoretically. By using frontier orbital analysis, AIM and LMOEDA, the strong acidic nature of the berlllium bonds was revealed. I recommend publication of this paper in this journal, subject to a minor revision: As pointed out by the GKS-EDA paper (Su, P.;* Jiang Z.; Chen, Z.; Wu, W. J. Phys. Chem. A, 2014, 118, 2531), the results of LMOEDA are sensitive to the choice of DFT functionals. Thus, the functional used in this paper for LMOEDA must be obviously shown in maintext.

Author Response

In this paper, the berlllium bonds in four beryllium derivatives of ortho-carboranes were explored theoretically. By using frontier orbital analysis, AIM and LMOEDA, the strong acidic nature of the berlllium bonds was revealed. I recommend publication of this paper in this journal, subject to a minor revision: As pointed out by the GKS-EDA paper (Su, P.;* Jiang Z.; Chen, Z.; Wu, W. J. Phys. Chem. A, 2014, 118, 2531), the results of LMOEDA are sensitive to the choice of DFT functionals. Thus, the functional used in this paper for LMOEDA must be obviously shown in maintext.

Authors’ reply:  Following the suggestion of the reviewer the DFT functional and basis set used for the LMOEDA calculations have been indicated in the computational method section (line 88 and 91-93).

Reviewer 2 Report

The authors present an interesting article on the structure and reactivity towards bases of a series of beryllium-containing cage compounds.  The article is well written and does an excellent job of presenting the results in an easy-to-understand manner.  I would suggest a few things for the authors to consider that might improve the quality of the paper:

  1. It is notable that complexes “B” have less symmetry than the others, especially with regards to the peculiar orientation of the bottom-most BH bond, both in the uncomplexed structure as well as the structures complexed with the bases.  I think it worth commenting on this and possibly offering a reason for the asymmetry.
  2. In the paragraph of lines 136-142, the authors discuss a measure of molecular strain based on the deviation of the Be atom from the plane defined by the boron atoms to which it is attached. The text suggests that the Be atom would prefer to be in the same plane as the B atoms.  Is this true?  If so, why does the Be atom deviate from the optimal coplanar situation?
  3. From the molecular graphs of the various complexes, it is interesting that the bond critical point is very close to the Be in each bond, and should be noted in the text. Do the authors have a reason for this observation?
  4. There are so many different energy partition methods, that I find it difficult to associate a particular physical meaning with the various terms used. While the meaning of the electrostatic term is common, it would be very helpful if the authors could include a very brief interpretation of the meaning of the exchange, polarization and dispersion components to aid the reader in interpreting the data.  Also, I have seen bar graphs used to clearly illustrate the differentiation of the components for a series of compounds (i.e., each compound has a single bar, broken down by colors according to the participation of each energy component).  This makes it easier to spot the significant differences in participation of each component for the different compounds.
  5. I found Table 6 in the supporting information to be easier to be more informative than Table 4 in the text. Perhaps just include Table 6 in the text?  Especially revealing is that the deformation energies for each complex is relatively constant even across the large range of bases.  Also, is there a change in the deviation of the Be atom from the plane of connected B atoms (see 2 above) upon complexation?
  6. Finally, it seems to me that the overriding reason for ready complexation of Lewis bases to these compounds is the electrophilicity of the exposed Be atom. Are the authors suggesting that it is the strain in the molecules that is responsible for the electrophilicity?  I’m not sure I find convincing evidence of this.  Is it not possible it is simply the surface area of the exposed Be atom that correlates with the electrophilicity?

This is an interesting paper that deserves publication, although I would prefer that the authors consider my comments above.

Author Response

  1. It is notable that complexes “B” have less symmetry than the others, especially with regards to the peculiar orientation of the bottom-most BH bond, both in the uncomplexed structure as well as the structures complexed with the bases.  I think it worth commenting on this and possibly offering a reason for the asymmetry.

Authors’ reply: The asymmetry of this molecule and its complexes is simply due to the diverse bonding patterns in the icosahedral cage combining Be, C and B. We do not think this property dictates the Lewis acid behavior of these compounds. In fact, compounds A, C and D are symmetric but they exhibit totally different electrostatic potentials, as illustrated in Table 2.

  1. In the paragraph of lines 136-142, the authors discuss a measure of molecular strain based on the deviation of the Be atom from the plane defined by the boron atoms to which it is attached. The text suggests that the Be atom would prefer to be in the same plane as the B atoms.  Is this true?  If so, why does the Be atom deviate from the optimal coplanar situation?

Authors’ reply: The local geometry at the Be atom is dictated by the atoms bonded to it. For instance, in BeF2, the FBeF angle is 180°, while in CO3Be the analogous OBeO angle is 98°. In the beryllium ortho-carboranes studied here, the beryllium atom is bonded to a 5 membered ring and this forces it to be out of the plane approx. 1 Å. The calculated distance in a system where the beryllium atom is bonded to a 6 membered ring of boron atoms (B6H6) is reduced to 0.64 Å, while when it is bonded to a 4 membered ring (B4H4) it increases to 1.24 Å. Thus, this distance clearly measures the strain associated to the Be atom. Beryllium would be coplanar only if the rest of the structure were allowed to relax while keeping the B-B, C-C and B-C bonds, which is not the case. In the end, the overall energy is a balance between all bonds: we gain more strain on beryllium but keep the cage. We have rewritten this paragraph to make clearer why there is a direct connection between strain at the Be atom and its electrophilicity.   

3.

  1. From the molecular graphs of the various complexes, it is interesting that the bond critical point is very close to the Be in each bond, and should be noted in the text. Do the authors have a reason for this observation?

Authors’ reply: We have indicated in the revised version this fact and propose an explanation (lines 172-173). This is the general behavior in any A-B bond, in which A ¹ B. If A is less electronegative than B the electron density accumulated around A is smaller than that accumulated around B and, as a consequence, the BCP is always closer to the less electronegative atom.

4.

  1. There are so many different energy partition methods, that I find it difficult to associate a particular physical meaning with the various terms used. While the meaning of the electrostatic term is common, it would be very helpful if the authors could include a very brief interpretation of the meaning of the exchange, polarization and dispersion components to aid the reader in interpreting the data.  Also, I have seen bar graphs used to clearly illustrate the differentiation of the components for a series of compounds (i.e., each compound has a single bar, broken down by colors according to the participation of each energy component).  This makes it easier to spot the significant differences in participation of each component for the different compounds.

Authors’ reply: The meaning of the different terms has been explained in the computational method section (lines 94-97).

            Following the suggestions of the reviewer a figure with the energy terms in the LMOEDA has been included in the Supporting Information (Fig. S5).

  5.        

  1. I found Table 6 in the supporting information to be easier to be more informative than Table 4 in the text. Perhaps just include Table 6 in the text?  Especially revealing is that the deformation energies for each complex is relatively constant even across the large range of bases.  Also, is there a change in the deviation of the Be atom from the plane of connected B atoms (see 2 above) upon complexation?

Authors’ reply: This is a very good suggestion, so we have replaced the initial Table 4 in the main text by Table 6 of the SI.  

6.

  1. Finally, it seems to me that the overriding reason for ready complexation of Lewis bases to these compounds is the electrophilicity of the exposed Be atom. Are the authors suggesting that it is the strain in the molecules that is responsible for the electrophilicity?  I’m not sure I find convincing evidence of this.  Is it not possible it is simply the surface area of the exposed Be atom that correlates with the electrophilicity?

Authors’ reply: Linear BeX2 molecules bend when they form complexes with Lewis bases. The deformation energy associated to this process is in general large (See Table 4). Strained systems, as those studied in the present article, exhibit already X-Be-X bent arrangements and therefore avoid much of the deformation energy and consequently became more acidic.